# Gut Microbiota, Probiotic Interventions, and Cognitive Function in the Elderly: A Review of Current Knowledge

**DOI:** 10.3390/nu13082514

**Published:** 2021-07-23

**Authors:** Agata Białecka-Dębek, Dominika Granda, Maria Karolina Szmidt, Dorota Zielińska

**Affiliations:** 1Department of Human Nutrition, Institute of Human Nutrition Sciences, Warsaw University of Life Sciences (SGGW), Nowoursynowska 159c, 02-776 Warsaw, Poland; dominika_granda@sggw.edu.pl (D.G.); maria_szmidt@sggw.edu.pl (M.K.S.); 2Department of Food Gastronomy and Food Hygiene, Institute of Human Nutrition Sciences, Warsaw University of Life Sciences (SGGW), Nowoursynowska 159c, 02-776 Warsaw, Poland; dorota_zielinska@sggw.edu.pl

**Keywords:** gut microbiota, microbiome, cognitive function, cognitive impairment, probiotics, the elderly

## Abstract

Changes in the composition and proportions of the gut microbiota may be associated with numerous diseases, including cognitive impairment. Over the recent years, the growing interest in this relation is observed, but there are still many unknowns, especially in the elderly. To the best of our knowledge, this is the first work that synthesizes and critically evaluates existing evidence on the possible association between human gut microbiota and cognitive function in the elderly. For this purpose, comprehensive literature searches were conducted using the electronic databases PubMed, Google Scholar, and ScienceDirect. The gut microbiota of cognitively healthy and impaired elderly people may differ in the diversity and abundance of individual taxes, but specific taxes cannot be identified. However, some tendencies to changing the Firmicutes/Bacteroidetes ratio can be identified. Currently, clinical trials involving probiotics, prebiotics, and synbiotics supplementation have shown that there are premises for the claim that these factors can improve cognitive functions, however there is no single intervention beneficial to the elderly population. More reliable evidence from large-scale, long-period RCT is needed. Despite proposing several potential mechanisms of the gut microbiota’s influence on the cognitive function impairment, prospective research on this topic is extremely difficult to conduct due to numerous confounding factors that may affect the gut microbiota. Heterogeneity of research outcomes impairs insight into these relations.

## 1. Introduction

Aging is one of the inevitable and progressive biological processes that leads to irreversible physiological and functional changes throughout the body. Undoubtedly, aging is associated with the deterioration of the body’s condition over the years. The functions of many organs and systems weaken. At the same time, the body’s ability to withstand physiological burdens, fight infection, and maintain homeostasis decreases [1,2]. The aging process also leads to changes in the nervous system and the brain, and thus also to changes in cognitive functioning. Brain aging can go in one of three directions: successful aging—which proceeds without changes in the cognitive function, normal aging—when there is a slight deterioration of cognitive functions with age, and cognitive aging—including mild cognitive impairment and clinical cognitive disorders affecting everyday functioning [3]. Dementia is a general term used to describe a clinical syndrome characterized by progressive decline in cognitive domains, including memory, language, executive and visuospatial function, behavior, and ability to perform activities of daily living. The most common cause of dementia is Alzheimer’s disease (AD), but the term includes also other forms such as vascular dementia, frontotemporal dementia, dementia with Lewy bodies, or even mixed forms. AD is an progressive and irreversible neurodegenerative disease and accounts for 60–80% of dementia cases [4]. Cognitive deficits between normal aging and dementia disorders are collectively called mild cognitive impairment (MCI), often considered an early stage of AD, but resulted from a variety of etiologies [5]. Cognitive impairment related to dementia affects not only the everyday functioning and quality of life of elderly individuals, but it also exerts immense social and economic impacts. Therefore, identifying interventions to prevent or reduce the risk of its onset is an urgent public health priority [6].

The human microbiota can be defined as complex microbial community which exceeds the number of cells of the host [7]. A human microbiome consists of more than 30 trillion microorganisms per person, including bacteria, fungi, and viruses. It is estimated that the human microbiome accounts for 1–3% of one’s body weight [8]. There is an excellent interpersonal variety in the microbiome composition, which is due to various factors like age, diet, socioeconomic status, medication, and many others, some still to be discovered [9,10]. It is said that the gastrointestinal microbiome—gut microbiota—is a dynamic and functional interface between the external environment and the human body [11]. Gut microbiota is composed of several species of microorganisms, including bacteria, archaea, yeast, and viruses. The predominant gut microbial phyla typical for a healthy human are Firmicutes, Bacteroidetes, Actinobacteria, Proteobacteria, Fusobacteria, and Verrucomicrobia, with the two phyla Firmicutes and Bacteroidetes representing 90% of gut microbiota [12]. The Firmicutes phylum is composed of more than 200 different genera such as *Lactobacillus*, *Bacillus*, *Clostridium*, *Enterococcus*, and *Ruminicoccus*. *Clostridium* genera represent 95% of the Firmicutes phyla. Bacteroidetes consist of predominant genera such as *Bacteroides* and *Prevotella*. The Actinobacteria phylum is proportionally less abundant and mainly represented by the *Bifidobacterium* genus [13]. Several changes during aging were observed in gut microbiota diversity and among core species, which may affect human health [14].

Both cognitive functioning and the microbiome are influenced by many modifiable and non-modifiable factors, including lifestyle and diet [11,15,16,17,18]. It is considered that changes in the composition and proportions of the gut microbiota may be associated with numerous diseases; however, it should be emphasized that it is currently at an early stage of research. Despite proposing several potential mechanisms of the gut microbiota’s influence on the development of diseases, prospective research on this topic is extremely difficult to conduct due to numerous confounding factors that may affect the gut microbiota [19].

The work presents a review of the data on the association between human gut microbiota and cognitive function in the elderly in accordance with recently published investigations. Despite increased interest in this area in recent years, there is a lack of original research concerning the elderly. To the best of our knowledge, this is the first review on gut microbiota and nutritional interventions in the context of cognitive decline in the elderly. We asked ourselves what has been confirmed in this regard so far. For this purpose, we have identified three indirect questions: Are there any differences in the gut microbiota between healthy elderly and those with cognitive impairment such as mild cognitive impairment (MCI), dementia, and AD? Can nutritional factors improve cognitive functions in the elderly due to gut microbiota modulation? What are the possible mechanisms behind gut microbiota and cognitive impairment?

## 2. Review Methodology

The review methodology can be divided into several phases (Figure 1). In the first review phase, the planning phase, we identified research areas that could shed light on the review question. Several inquiry domains were deemed relevant, including research on the elderly with MCI, dementia, and AD.

The next phase of the review was the search for research and can be divided into two separate phases. In the first search phase, we used many keywords while looking for useful articles, among which were “microbiota”, “microbiome”, “gut microbiota“, “dysbiosis”, “mild cognitive impairment”, “cognitive function”, “dementia”, “Alzheimer’s Disease”, “microbiota-gut-brain axis”, “elderly”, and “older adults” (*n* = 9560). In the searching process, we used many databases, including PubMed, Google Scholar, ScienceDirect, Scopus, as well as Cochrane. Three authors did the research independently to avoid the omission of essential studies in this field. In the second search phase, we deepened our search for intervention proposed to modify gut microbiota that improves cognitive functions in the elderly, and for this purpose, new keywords were used, such as “probiotics”, “prebiotics“, “psychobiotics”, “postbiotics”, “synbiotic”, “lactobacillus”, “bifidobacterium”, and “diet intervention” (*n* = 2724).

In the selection phase, titles and abstracts of articles retrieved during the search phase were screened, and relevant articles were selected. When selecting the publications, we were guided by the following criteria: (1) if possible, recently published studies; (2) studies published in reviewed journals; and (3) studies conducted with the participation of the elderly.

In the evaluation phase, the conceptual contributions, strengths, and limitations of selected articles were noted and critically evaluated in light of the review questions. This phase included studies concerning microbiome composition in cognition impairment (*n* = 6) and randomized controlled trials (RCT) studies concerning probiotic, prebiotic, and synbiotic intervention on cognition function (*n* = 12). It was relevant for this review to present the research results in a broad perspective, taking into account several larger issues (review of observational and interventional studies, the relationship between diet, the microbiome and cognitive disorders in the elderly, physiological mechanisms), but due to the small number of original studies and their heterogeneity, it was not possible to provide a systematic review. We aimed to provide a comprehensive view, showing the current state of knowledge, point out the errors and gaps, and the advancement of the research.

Due to the complexity of the analyzed issue and a great variety of terms, we provided a glossary of frequently used terms (Table 1).

## 3. How Does the Gut Microbiota Change with Aging?

The microbiome composition differs even among healthy people, as there are multiple interindividual differences. The density and composition of the microbiota are affected by chemical, nutritional, and immunological gradients along the gut [11]. In the small intestine, there are typically high levels of acids, oxygen, and antimicrobials, and a short transit time, which is why it is mainly colonized by facultative anaerobes like *Proteobacteria* and *Lactobacillales*, but in the more distal area, gut microbiota becomes more diverse and approaches that of the colon [8]. Although to remain healthy, not only abundance, but also the microbiome’s diversity, seem crucial, some examples of bacteria classification are potentially beneficial (*Bifidobacterium*, *Lactobacillus*, *Eubacterium*, and *Fusobacterium*) and potentially harmful (*Staphylococcus*, some of *Clostridia* and *Proteus*). Pathogenic bacteria do not occur extensively in a healthy human microbiota [28]. However, most of the gut microbes were mainly commensals (normal microbiota), which play a pivotal role in human health modulation, e.g., immune function, metabolic function, as well as neurobehavioural traits [14]. A decline in gut microbiota diversity is one of the most common changes during the aging process. Core species tend to decrease with the increased abundance of subdominant species [29]. The numbers of Firmicutes, *Faecalibacterium prausnitzii*, and *Clostridium* are significantly reduced in people ≥65 years old [30]. Researchers also report a significant reduction of *Lactobacillus* and *Bifidobacterium*, which is related to lower levels of its metabolites: short-chain fatty acids (SCFAs) and γ-aminobutyric acid (GABA), which play an essential role in the nervous system, for example, in the regulation of neurotransmitter secretion. The only one of the dominant species of *Bifidobacterium* genus found in old age is *B. adolescentis*, or phenotypically close *B. angulatum* and *B. longum* in comparison in adult microbiota, where 4–5 species of *Bifidobacterium* are present [31,32]. Other authors also reported decreased *Bacteroides* and *Enterobacteriaceae* levels in centenarians compared to younger adults [33,34]. The increase of proteolytic bacteria, such as *Fusobacteria, Propionibacteria*, and *Clostridia*, was found in the intestinal microbiota of elderly people leading to the development of putrefactive processes, especially in patients with post-antibiotic therapy. Furthermore, an increased number of proinflammatory *Enterobacteriaceae*, *Streptococcus*, *Staphylococcus*, and yeast cells were found, which may be associated with an elevated level of serum antibodies to commensal microbiota, such as *Escherichia coli* and *Enterococcus faecalis* [32,35]. Population differences should also be considered while analyzing microbiota changing data because it also influences microbiome composition, as there are different environmental conditions, level of hygiene, eating habits, and many more factors [29]. There are also differences in the gut microbiota composition in different geographic regions of the world which is most often explained by the diversity in the eating habits—*Bacteroides* enterotype is more common in Western countries (with high fat and protein intake) while *Prevotella* enterotype is more abundant in non-Western countries with high fiber consumption. The population of Africa is characterized by a high gut microbiota diversity (dominated by Actinobacteria, Bacteroidetes, Firmicutes, Proteobacteria, and Spirochaetes), while Europeans and Americans gut microbiota is enriched in Firmicutes, Actinobacteria, Verrucomicrobia, and Bacteroidetes [36].

It is not clear whether it is the aging process that influences gut microbiota changes or a combination of various factors related to the elderly. Gut microbiota changes might be associated with a general deterioration of health, the occurrence of chronic diseases, multi-drug use, including antibiotics and non-steroidal anti-inflammatory drugs, but also with changing lifestyle habits [15]. For example, antibiotics are now considered in two ways in influencing gut microbiota in the pathogenesis of AD: on the one hand, overuse of antibiotics may lead to decreased diversity and cause dysbiosis, but on the other hand, there is also the possibility of the use of antibiotics as therapeutic manipulation of gut microbiota [37]. Older people often eat a monotonous diet as a result of losing a partner due to death or divorce and living alone, repeatedly giving up preparing wholesome meals due to a lack of motivation to cook every day [38]. Furthermore, problems with dentition lead to choosing only known products with a certain hardness and structure, and not using raw vegetables and fruits or nuts [39]. As mentioned above, changing nutritional habits are very often connected with malnutrition, which is present in 5–30% of community-dwelling older adults, with higher numbers reaching 60% in acute care settings [40,41]. Not only in the oral cavity, but also in further parts of the gastrointestinal tract, there are some important changes connected with aging, which may influence the gut microbiota composition: for example, the compliance of gastric and rectal is lowered, which results in prolonged oro-caecal and colonic transit time [15]. Older people often limit their physical activity due to mobility limitations, fear of injury, or lack of motivation [42,43]. All of these factors can indirectly influence the composition and diversity of the gut microbiota and therefore affects human health [44,45,46].

## 4. Gut Microbiota in Cognitive Disorders—Is There a Difference?

The evidence regarding gut microbiota differences in patients with various kinds of cognitive decline is insufficient to perform a deep analysis and draw unequivocal conclusions. We found only single studies that compared the composition and abundance of the gut microbiota in healthy older people and people with cognitive impairment—the results are shown in Table 2 and the methodological details are presented in Table 3.

Saji et al. studied 128 patients of a memory clinic, of which 94 were classified as non-demented and 34 as demented based on the Mini-Mental State Examination (MMSE) score and Clinical Dementia Rating Score. The authors showed a significantly lower percentage of *Bacteroides* in the dementia group than non-dementia and slightly, non-statistically significantly more frequent *Lactobacillus* and *Bifidobacterium* in the dementia group, which was unexpected because of its pro-health properties [47,48]. Another surprising result was that the Firmicutes/Bacteroidetes ratio was significantly higher in the dementia group. Components of the gut microbiota were independently associated with dementia, even more strongly than traditional dementia biomarkers. However, it should be pointed out that the authors included patients with mild cognitive decline in the non-demented group, which might have influenced the results [49]. The authors also provided another analysis with the participation of some patients from the previously described study, but this time they excluded patients with dementia and divided the 82 remaining patients into two groups: mild cognitive decline (MCI) and cognitively healthy group. Patients with MCI had a higher prevalence of *Bacteroides*; a multivariable logistic regression analysis showed that a greater prevalence of this genus was independently associated with MCI. However, these two studies are of cross-sectional design, so it is impossible to conclude about cause-and-effect dependencies. It should also be underlined that these results were only obtained in a relatively small sample of the Japanese population, and there is a great need to conduct further research on different and more representative populations [50].

Decreased gut microbiota diversity and changes in its composition, including decreased Firmicutes, increased Bacteroidetes, and decreased *Bifidobacterium* in patients with AD, were found in a study conducted with the participation of 25 AD and 94 non-demented people from the American population [51]. Furthermore, researchers from Italy, in their study that included 83 patients (10 cognitively healthy amyloid negative controls, 40 cognitively impaired amyloid positive, and 33 cognitively impaired amyloid-negative), showed that patients with cognitive impairment and amyloidosis had a lower abundance of anti-inflammatory taxa *Eubacterium rectale* and higher levels of the proinflammatory taxes *Escherichia/Shigella* in gut microbiota [52]. Similar results concerning lower counts of *Eubacterium rectale* in demented patients have recently been obtained in the Austrian population [53]. The authors also noted a significantly lower abundance of uncultured *Lachnospiraceae* sp. and *Lachnospiraceae* NK4A136 in the group of 23 demented patients, which was also proved in the group of AD patients and patients with amnestic MCI when compared to healthy controls. The authors provided an extensive analysis on different classes and families of gut microbiota and revealed many statistically significant differences in the gut microbiota composition in patients with AD and healthy controls (detailed results are presented in Table 2) [54].

In contrast, no significant differences in microbiome diversity were shown in mildly cognitively impaired subjects compared to cognitively healthy subjects in a small group of 17 patients (11 MCI vs. cognitively healthy) [55]. However, the authors detected several unique microbial signatures in subjects with MCI.

Recently, scientific evidence highlighted the role of oral microbiota in influencing brain functions. Numerous studies have shown that periodontal issues are also associated with neurodegenerative and cognitive decline [56,57,58]. It was found that *Porphyromonas gingivalis*, a bacterium associated with periodontitis, has been implicated in dementia. It has been also suggested that the bacteria is capable of modulating the gut microbiota [59].

In summary, small-sample studies have shown that the gut microbiota of cognitively healthy and impaired elderly people may differ in the diversity and abundance of individual taxes, but due to the small number of studies and divergent results, specific taxes cannot be clearly identified. However, some tendencies to changing the Firmicutes/Bacteroidetes ratio can be identified. Gut microbiota diversity seems to be one of the crucial factors in retaining a healthy microbiome in older age; however, its assessment is connected with multiple methodological issues.

## 5. Is There a Link Between Diet, Cognitive Function, and Gut Microbiota in the Elderly?

Available reports on the existing differences in the composition and diversity of gut microbiota, which depend on cognitive functioning, give rise to the question of whether it is possible to manipulate the gut microbiota to improve cognitive functions or to reduce the risk of cognitive impairment in the elderly. Multiple environmental and host factors play an important role in shaping gut microbiota composition and function. Within environmental factors, diet is considered as a key factor [60], both as short-term changes [61], as well as long-term dietary patterns and habitual intake [62]. Due to the lack of long-term human studies, it cannot be determined whether prolonged dietary changes can induce permanent alterations in the gut microbiota, but it is possible that habitual diets have a greater influence than acute dietary changes [63].

Several popular diets, including the Western, vegetarian, vegan, ketogenic diet, and elimination diets such as low-FODMAP and gluten-free diet, have been studied for their ability to modulate the gut microbiota diversity [45,64]. The Mediterranean Diet is one of the most discussed nutritional models in the literature in the context of cognitive enhancement. This diet is considered one of the healthiest in the world because of its documented association with reduced morbidity and mortality due to many chronic diseases [65]. Studies showed that adherence to the Mediterranean diet is associated with a reduced risk of stroke, depression, and cognitive impairment [66]. This is also confirmed by the meta-analysis of prospective studies by [67] Singh et al., which showed that people who adhere to the Mediterranean diet, to a greater extent, had a lower risk of MCI and AD, as well as the progression of MCI to dementia [67]. Moreover, studies have observed a beneficial effect of the Mediterranean diet on the cognitive functioning of healthy elderly people [68]. Recent data demonstrate that adherence to the Mediterranean diet promotes changes in the gut microbiota diversity and richness, by increasing Bacteroidetes and Firmicutes, changing the ratio of Bacteroidetes/Firmicutes, and increasing total levels of fecal SCFAs [69,70,71]. However, only a few studies have evaluated the correlation between the three variables: dietary patterns, gut microbiota, and cognitive function in older adults. Adherence to the Mediterranean diet for 12 months was associated with specific microbiome alterations in 612 elderly subjects. Taxa along with adherence to the diet were positively associated with several markers of lower frailty and improved cognitive function, and negatively associated with inflammatory markers including C-reactive protein and interleukin-17 [72]. A randomized, double-blind, crossover, single-center pilot study of Mediterranean-ketogenic diet (MMKD) six-week intervention on 17 elderly subjects (11 MCI and six cognitively normal) suggested that MMKD can modulate the gut microbiome, SCFAs level, and AD biomarkers in cerebrospinal fluid, including the deposition of β-amyloid (Aβ)-40 and Aβ-42 [55].

Some nutrients have been shown to affect the composition of the gut microbiota as well as cognitive function (Table 4). For example, omega-3, polyphenols, and vitamin D appear to have the potential to confer health benefits via modulation of the gut microbiota [45]. A little more is known about the impact of nutrients on cognitive function than the gut microbiota in the elderly. Observational studies indicate the role of poor nutritional status (e.g., low vitamin D status, low plasma level of antioxidant) in poorer cognition and an increased risk of cognitive impairment, but RCT studies do not confirm that the use of dietary supplements can help maintain cognitive health. The RCT studies showed that fiber and vitamin D influence the composition of the intestinal microbiome. Both, in the case of cognitive function and gut microbiota, the adverse effect is noted for saturated fatty acids. However, to the best of our knowledge, there are no clinical trials that would link nutrients, gut microbiota, and cognitive functions. In this review, we focus only on probiotic interventions in the elderly and their effects on cognition by modifying the gut microbiota.

## 6. Can Administering Probiotic or Prebiotics Supplementation Improve Cognitive Functions in the Elderly?

It has been noted that the elderly buy probiotic supplementation expecting to improve their health [87]; however, probiotic effects on the cognitive function in this particular group have not been studied well. The “psychobiotics” theory indicates that probiotics have a potential, positive effect on mental health, but more research on this relation is warranted [88]. The health effects of microbiota-driven therapy (pre-, pro-, and synbiotics) on the elderly include changes in the composition and activity of the intestinal microbiota, especially by promoting the growth of bifidobacteria and lactobacilli [89]. However, it should be emphasized that only limited studies were conducted specifically on the elderly.

The results of the meta-analysis by Lv et al. showed that probiotics supplementation enhanced cognitive function in human and animal studies, and the effects on cognitively impaired individuals were greater than those on healthy ones. Moreover, the authors showed that a duration of less than 12 weeks and a single strain of probiotics were more effective in human studies. This meta-analysis included seven human studies, where three studies included subjects diagnosed with AD, one with healthy elderly, and three with cognitive impairment participants (fibromyalgia, major depressive disorder, minimal hepatic encephalopathy) where the average age was under 60 years old [90].

To the best of our knowledge, only a few studies have assessed this effect both among healthy elderly and those with cognitive impairment. The methodological details of the discussed studies are presented in Table 5. The studies with healthy older adults give different results. The lack of beneficial effect in cognitively healthy subjects was noted in the study by Benton et al., which unexpectedly showed that after 21 days of ingestion of a probiotic milk drink containing *L. casei* Shirota or placebo, memory abilities were slightly worse in the probiotic group [91]. On the other hand, other studies involving healthy individuals indicate potentially beneficial effects of probiotics. The administration of *Lactobacillus helveticus—*fermented milk drink—for 8 weeks in middle-aged healthy adults [92] and 12 weeks in healthy elderly people [93] improved the cognitive function compared to the placebo group. Moreover, Kim et al. demonstrated that probiotics (*Bifidobacterium bifidum* BGN4 and *Bifidobacterium longum* BORI) consumed for 12 weeks improved cognitive and mental health as well as changed the gut microbiota composition in healthy community-dwelling elderly [94]. Inoue et al. demonstrated that combined probiotic bifidobacteria supplementation and moderate resistance training may improve the mental condition, body weight, and bowel movement frequency in healthy elderly subjects [95].

The results of studies involving participants with cognitive impairment are the subject of several meta-analyses. Den et al. indicated that probiotics significantly improved cognitive performance in AD and MCI patients (SMD = 0.37; 95% CI, 0.14 to 0.61). This meta-analysis included five randomized controlled trials (RCT), where three studies included subjects diagnosed with AD and two with MCI [96]. However, another meta-analysis, which included the same data but considered only individuals with AD (3 RCT), showed no beneficial effect of probiotic supplementation on cognitive function (SMD = 0.56; 95%CI: −0.06 to 1.18) [97].

The discrepancy in these meta-analyses results suggests that there is too few data for unequivocal conclusions to be drawn. It needs to be highlighted that the number of studies assessing the effect of the probiotic intervention on cognitive function in elderly people is limited. Many unknowns still require clarification, and many parameters need to be controlled both from the probiotic and host sides when planning studies. The illustration of this issues is presented in Figure 2.

From the probiotic’s side:The health-promoting effect of a probiotic depends on the strain. Some strains show a positive effect on cognitive function, and some do not. The research uses either single strain or multiple strain probiotics. Lv et al. observed that a single strain of probiotics was more effective in human studies [90]. However, as previously mentioned, diversity may be crucial for the cognitive health of the elderly, but most studies have not assessed changes in the microbiota after administering probiotic supplementation.There is not enough evidence to provide information on dose–response functions associated with probiotics. Most studies have not compared the different doses. Doses that ranged from 10^8^ to 10^11^ were used most often, but more reliable evidence from various dosages is needed, especially outside these common doses [96].There are also many unknowns about the duration of probiotics ingestion. The most common intervention duration of the studies was 12 weeks. Some authors emphasize that the intervention could have been too short [94,99]. Lv et al. suggest that a duration of less than 12 weeks was more effective, but these studies included data from middle-aged adults [90]. There is a lack of long-term studies especially in the group of elderly.

From the host’s side:The success of the intervention could be modified by the hosts’ diet and lifestyle, age, sex, geographic region, concomitant disease, antibiotic exposure, and baseline microbiota composition [105,106,107]. All these factors should be controlled. Still, the studies did not assess the effectiveness of probiotic intake through gut microbiota composition. Most studies did not consider the baseline gut microbiota characteristics of included individuals. Despite the influence of diet on the gut microbiome, the majority of studies contain little or no analysis of dietary intake. There is a need of conducting more well-controlled longitudinal studies and randomized controlled trials that may isolate the impact of specific changes in dietary intake on gut microbiota [108].The level of cognitive functioning can make a difference. There are no studies that would compare people with different levels of cognitive functioning. Kobayashi found a beneficial effect of probiotics in the low-score subgroup but not in the high-score subgroup (indicating favorable cognitive performance), so they suggest comparing the results between MCI or early dementia and cognitively normal individuals [99]. Lv et al. showed that the effects of probiotics supplementation were greater in the case of cognitively impaired individuals than those on healthy ones, but they only included one study with cognitively healthy people [90]. What is more, the current RCT, including patients with cognitive impairment, are mainly focused on inflammatory and oxidative biomarkers rather than cognitive function, so they used MMSE or TYM, which are used for screening rather than cognitive assessment [100,101,102].The side-effects of probiotics intervention should be considered alongside the observed benefits [109]. Hibberd et al. reported that *Lactobacillus rhamnosus* GG ATCC 53,103 (LGG) (1 × 10^10^ CFU) is safe and well-tolerated in healthy adults aged 65 years and older [110]. Similarly, the results of Hwang et al. suggest that *Lactobacillus plantarum* C29-fermented soybean can be safely administered to enhance cognitive function in individuals with MCI. Adverse events observed were stomach aches, headaches, gastritis, erectile dysfunction, and seborrheic dermatitis, all of which were classified as a mild adverse event [98].

Only single RCT studies on the effects of prebiotics and synbiotics supplementation on cognition in the elderly are available (Table 5). Buigues et al. evaluated whether the regular intake of prebiotic can improve frailty criteria, functional status, and response of the immune system in elderly people affected by the frailty syndrome. They assessed the effect on functional and cognitive behavior as well as sleep quality, but no significant effects were observed [103], whereas Louzada and Ribeiro investigated the effect of a synbiotic on symptoms of brain disorders and inflammation in the elderly, and found weak effects of synbiotic on depressive symptoms and more optimistic effects on cognition in apparently healthy elderly [104].

## 7. What Is the Link Between the Microbiota–Gut–Brain Axis and Cognitive Function?

Recent studies have shown that the gut microbiota is involved in the neurodevelopment and diverse brain functions through regulating the gut–brain axis. The term “microbiota–gut–brain axis” (broadened from “gut–brain axis”) [111] represents the bidirectional communication between the central nervous system (CNS) and the gastrointestinal tract [111]. The communication between CNS and the enteric nervous system is multichannel and involves neural, immune, endocrine, and metabolic signaling mechanisms [112,113]. As we presented above, it is still discussed if dysbiosis of gut microbiota, as well as probiotics and prebiotic supplements, may influence cognitive function, especially since the mechanism underlying between the microbiota–gut–brain axis and cognitive function is not fully understood.

Human gut microbiota produces various metabolites that influence the nervous system directly like noradrenaline, serotonin, acetylcholine, SCFAs, and GABA produced by *Lactobacillus* and *Bifidobacterium* or dopamine produced by *Escherichia* and *Bacillus*. An essential feature of gut microbiota metabolites is modulating both microbe-microbe and microbe–host interactions, which all impact human health [8,45]. In terms of interbacterial communication, gut bacteria produce many signaling molecules, known as quorum sensors, which may influence apoptosis, growth, and bacterial homeostasis. Other metabolites influence multiple host processes via communication with the central nervous system (like GABA). Some gut microbiota components can produce vitamins necessary for proper immune functioning like K and B group vitamins [8,31].

In recent years, the number of studies that focus on explaining this mechanism has been increasing, but most of the research has been done within animal models. In a scoping review, Gao et al. distinguished five main pathways through which the microbiota–gut–brain axis may modify cognitive function [114]. First, gut microbiota dysbiosis has been shown to have an impact on the function of the hypothalamic–pituitary–adrenal (HPA) axis, which regulates cognitive function. The second pathway is the neuronal regulation pathway, where gut microbiota communicates with the CNS and autonomic nervous system (ANS) via vagus nerve stimulation [114,115]. The other pathway is based on microbiota and its metabolites that can stimulate the increased release of inflammatory markers, interacting with the blood–brain barrier. Fourth, the gut microbiota products, such as neurotransmitters or hormones, have been shown to modify CNS function. The last pathway that can explain gut microbiota’s role in the regulation of cognitive function is its induction of modification of myelination, myelin plasticity, microRNA expression in the prefrontal cortex, which plays an important role in planning and decision making. As the authors suggest, the mechanism through which gut microbiota modifies cognitive function is still not well understood; it is likely a combination of presented mechanisms [114].

## 8. What Are the Possible Mechanisms behind Gut Microbiota and Dementia?

Although many pathways are considered in the link between gut microbiota and cognitive function, little is known about detailed mechanisms among the elderly with dementia. Knowledge about the physiopathology role of the gut microbiota is much more established in other neuropsychiatric disorders (e.g., Parkinson’s disease) [116].

However, the pathomechanism of dementia remains not fully understood; the inflammatory process appears in most of the hypotheses as having a key role in cognitive decline. It is well established that aging is associated with inflammation [117], and this state is often referred to as “inflammaging” (chronic state of inflammation) [117]. Furthermore, as described previously, with aging, the diversity of the gut microbiome decreases, and recent studies have shown that gut microbiota may promote chronic inflammation [116]. In view that the main risk factor of dementia is aging, which is associated with the inflammation process, which in turn increases gut permeability, bacterial translocation, and others, it is hypothesized that the microbiota–gut–brain axis may explain the link between aging, dementia, and inflammation [54,117].

Recent studies also indicate increasing attention to microbiome-associated metabolites, which could alone affect and modulate cognitive function, and thus dementia [118]. Saji et al. (2020) hypothesized that higher concentrations of fecal metabolites of the gut microbiome might be associated with the presence of dementia, independently of the other risk factors for dementia and dysregulation of the gut microbiome [118]. The authors indicated that the concentrations of such metabolites as ammonia, formic acid, iso-butyric acid, is-valeric acid, phenol, and p-cresol were significantly higher among patients with dementia compared to non-demented controls. Furthermore, each one standard deviation increment in the fecal ammonia concentration was associated with a 1.6-fold risk for the presence of dementia. This study even showed a predictive potential of microbiome-associated metabolites in the diagnosis of dementia. A higher concentration of fecal ammonia and lactic acid indicated the presence of dementia and had a similar predictive value as traditional biomarkers for dementia [118].

Furthermore, as described previously, the gut–brain–axis affects several functions in the brain, such as the regulation of the HPA axis, rule actions in the periphery and central nervous system, producing neurotransmitter, endocrine hormones, immunomodulators on neuropeptides, which can all modulate dementia, but there is little research among elderly [32,116,118].

Much more is known about the mechanisms behind gut microbiota and AD (Figure 3). The most widely accepted explanation for the pathogenesis of AD is the accumulation of amyloid plaques (extracellular) and neurofibrillary tangles (intracellular) [119]. Furthermore, neuroinflammation, mitochondrial dysfunction, cerebral hypoperfusion, and impairment in the calcium balance are considered in the pathogenesis of AD [119,120]. As there are still major gaps in the understanding of the pathogenesis of AD, it is recognized as an interaction between genetic and environmental factors [121]. Over the past decade, it has been hypothesized that AD may be associated with gut microbiota dysbiosis. Microbiota dysbiosis may lead to increased permeability of the gut and blood–brain barrier, secretion of large amounts of microbial amyloids, lipopolysaccharides, neurotransmitters, and neurotoxins. Furthermore, imbalances in gut microbiota can induce inflammation, neuroinflammation, and oxidative stress.

In 2017, the new term “mapranosis” was proposed, representing microbiota-associated proteopathy and neuroinflammation [122]. One of the possible mechanisms underlying the link between gut microbiota and AD is the promotion of amyloid formation by human proteins [122]. The gut microbiota produces a significant number of amyloids [122], although microbial amyloids differ from the human CNS amyloids (they have similarities in tertiary structure) [123,124,125]. Examples of such bacteria are *Escherichia*, *Bacillus subtilis*, *Salmonella enterica*, and *Salmonella*
*typhimurium* [122,126,127]. Bacterial amyloid proteins in the gut may enhance the immune system to neuronal amyloid production in the brain. Bacterial amyloids may act as prion proteins, causing the amyloidogenic protein to adopt a pathogenic β-structure [128]. Furthermore, microbial amyloids may enhance the inflammatory response to endogenous neuronal amyloids [122].

Another possible mechanism proposes that lipopolysaccharides (LPS) may play a role in AD. Recently, *E. coli* and *B. fragilis* have been shown to intensely produce proinflammatory lipopolysaccharides (LPS), which can access brain compartments by crossing the gastrointestinal tract (GI tract) and blood–brain barrier [129]. The LPS presence in the brain (hippocampus and neocortex) has been detected post-mortem in AD patients [129]. In addition, AD patients had a 3-fold higher plasma concentration of LPS compared to healthy people [129]. Furthermore, recent intervention animal studies showed that bacterial LPS may induce a β-structure of prion amyloids and promote amyloid fibrillogenesis. There is also the hypothesis of a vicious cycle that is responsible for the progression of AD [130]. The hypothesis assumes that LPS, by acting on leukocyte and microglial TLR4-CD14/TLR2 receptors, increases Aβ levels (via the increased level of cytokines), while Aβ1–42 is an agonist for TLR4 receptors [130]. An increased level of cytokines also leads to damage of oligodendrocytes and produces myelin injury in the AD brain [130].

Over the years, inflammation has been implicated in AD pathogenesis, while recently, one of the sources for this inflammation has been indicated in GI. One of the markers of intestinal inflammation is calprotectin concentration in the stool [131]. Calprotectin is a heterodimer of proteins S100A8 and S100A9, which can form amyloid oligomers and fibrils resembling amyloid polypeptides, such as amyloid β and α-syn [131,132]. Kowalski et al. hypothesized that it is possible that intestinal calprotectin may contribute to amyloid fibril formation both in the gut or directly in the brain [128]. Recent studies showed that 70% of AD patients had a significantly elevated level of fecal calprotectin [133] compared to controls. It was also shown that S100A9 expression was increased in the brains of AD mice and AD patients [134]. These results are in line with a recent study, which indicated a significantly increased S100A9 level in the cerebrospinal fluid of AD patients [135]. The neuroinflammatory state has been implicated in AD and proposed to facilitate neurodegeneration [136]. The immune system’s hyperstimulation may be harmful to neurons and may lead to neuroinflammation when neurons release substances that sustain the inflammatory process and the immune response [37]. Cattaneo et al. showed that compared to controls, in patients with cognitive impairment and brain amyloidosis, an increase in the abundance of a proinflammatory *Escherichia/Shigella* and a reduction in an anti-inflammatory *E. rectale* were observed simultaneously [52]. Furthermore, a significant positive correlation was observed between pro-inflammatory cytokines IL-1β, NLRP3, and CXCL2 with an abundance of *Escherichia/Shigella* and a negative correlation with the *E. rectale* [52]. Furthermore, the production of other proinflammatory cytokines may be induced indirectly by gut microbiota. For example, proinflammatory taxons that can secrete the previously mentioned amyloid peptide curli indirectly activate the production of IL-6, IL-1β [137], IL-17A, and IL-22 [138]. Some of the cytokines are able to cross the blood–brain barrier and lead to neuroinflammation and neurodegeneration [139]. Furthermore, gut microbiota can modulate the circulating concentration of tryptophan (Trp), which in 95% is metabolized through the kynurenine pathway (KP). KP results in the production of neuroprotective kynurenic acid (KYN), 3-hydroxyanthranilic acid (3-HAA) (with antioxidant properties), as well as neurotoxic metabolites such as metabolites 3-hydroxykynurenine (3-HK) and quinolinic acid (QUIN) [140,141,142]. The observations linking the KP and AD include decreased tryptophan and 3-HAA concentrations in AD plasma, increased KYN/Trp ratio and 3-HK in serum of AD patients compared to elderly controls, and accumulation of QUIN in the hippocampus of AD patients [141,143,144,145]. Gut microbiota dysbiosis alters tryptophan metabolism, resulting in changes in the balance of the neuroprotective/neurotoxic kynurenine pathway, which seems to play an important role in AD pathology by leading to neuroinflammation [140].

Furthermore, the role of gut microbiota in regulating the oxidative state of the CNS may not be forgotten. Oxidative stress plays a significant role in AD pathology (e.g., enhances A*β* deposition), while antioxidant treatment effects in the retardation of the progression of AD [146]. In healthy individuals, gut microbiota may regulate the oxidative state via the production of various metabolites (SCFA, polyphenols, and absorbable vitamins) and enzymes (superoxide dismutase and catalase) [147,148,149], whereas if microbiota dysbiosis occurs, the ability to interfere with the level of reactive oxygen species (ROS) fades and the antioxidant defense system weakens [150]. Several studies showed increased peroxidation in AD patients and decreased levels of antioxidant molecules and activity of antioxidant enzymes. Shao et al. identified oxidative modifications to DNA and RNA via high levels of 8-hydroxy-2′-deoxyguanosine (8-OHdG) in the brain [151]. While differences in the activity of superoxide dismutase (SOD), glutathione peroxidase (GPx), 8-oxyguanine DNA glycosylase-1 (OGG1), and catalase (CAT), and levels of glutathione (GSH) and total antioxidant capacity (TAC) may indicate decreased activity of antioxidant enzyme systems among AD patients [151,152,153]. In addition, many researchers showed that probiotics demonstrate antioxidant abilities and could exert antioxidant capacity in different ways [154]. Tamtaji et al. showed that probiotic and selenium co-supplementation with selenium was associated with a significant increase in total antioxidant capacity and decreased levels of high sensitivity C-reactive protein (hs-CRP) among AD patients [102]. Den et al., in a meta-analysis of randomized controlled trials among adults with Alzheimer’s or mild cognitive impairment, showed a significant improvement in cognition, while a significant decrease in levels of malondialdehyde (MDA) and hs-CRP between probiotics and the control group. No significant differences in TAC and GSH were found between groups [96]. Recent studies also indicate that prebiotic and synbiotic supplementation may also have significant ability to improve metabolic activity and improve cognition in AD patients. Synbiotics showed to increase bioavailability of microbially produced antioxidant metabolites, enhancing activity of antioxidant systems and improving cognitive function among AD patients [155,156]. Within possible mechanisms behind AD, there is also one related to the secretory function of gut microbiota, which may modulate brain plasticity, affect cognitive function and develop AD via other than previously mentioned neurotransmitters and neuromodulators (e.g., serotonin, GABA, and histamine), or neurotoxic metabolites (e.g., D-lactic acid and ammonia) [38,113,131].

To sum up, the gut microbiota may secrete significant amounts of amyloids and lipopolysaccharides, which can enhance inflammatory response to endogenous neuronal amyloids and increase the proinflammatory production of cytokines associated with the pathogenesis of AD. Furthermore, microbiota dysbiosis may lead to increased permeability of the gut and blood–brain barrier and modulation of the kynurenine pathway. In this position, transmitters and neurotoxic substances can easily enter the brain, affect nerve function, and lead to the development or progression of cognitive impairment and AD. All of the mentioned possible mechanisms may play a part simultaneously. Naturally, AD is probably associated with multiple pathomechanisms and etiologies; gut microbiota appears as a significant part of the pathophysiologic process. A significant limitation of most studies investigating microbiota’s role in AD pathomechanism is that they are animal studies or cross-sectional studies. This causes difficulties in establishing an association between gut microbiota and the development/progression of AD. Additional case-control and prospective studies (including post-mortem studies) among patients with AD are required to better understand the link between gut microbiota and cognitive function/disease.

## 9. Conclusions

Over the recent years, growing interest in the relation between gut microbiota and cognitive function, especially in the elderly is observed, but there are still major gaps in our understanding of these interactions. Gut microbiota of cognitively healthy and impaired elderly people may differ in the diversity and abundance of individual taxes, but due to the small number of studies and divergent results, only some tendency to changing the Firmicutes/Bacteroidetes ratio can be identify. There will be interesting to find some specific taxes of microorganisms that could be serve as a marker of early stage of dementia. Such factors will be extremely important in preventing cognitive impairment development.

Currently, clinical trials involving dietary interventions and supplementation with probiotics, prebiotics, and synbiotics have shown that there are premises for the claim that these factors can improve cognitive functions, however it was not shown that there is a single intervention beneficial to the elderly population. More reliable evidence from large-scale, long-period RCT is needed. Previous heterogeneity of research outcomes impairs further insight into these relations. It is difficult to spot any unambiguous dependencies due to methodological difficulties and a multitude of factors influencing both cognitive functions and gut microbiota. The effects of probiotics supplementation seem to be greater on cognitively impaired individuals than those on healthy ones, but more proof is needed in this area. Despite that we know that the health-promoting effect of a probiotics depends on the strain, doses and duration of probiotics ingestion, there is still lack of studies compared different intervention, e.g., different genus or species, different doses. It is also unclear whether the effect will persist after the end of the intervention and for how long. Future studies should also consider the baseline of microbiota composition and dietary intake of included individuals. The results of such studies should provide scientific clues for the rational design of dietary intervention specific to groups of older adults, with particular emphasis on the safety of using probiotic and prebiotic products. These findings are of public health importance because of the rapidly aging population and increasing cases of cognitive function diseases. Whether gut microbiota, probiotics/synbiotics may improve or slow down the cognitive function decline in the elderly needs further study. What is more, it seems that personalized nutrition strategies will be particular importance in the future, which is already noticeable in growing interest in this area.

## Figures and Tables

**Figure 1 nutrients-13-02514-f001:**
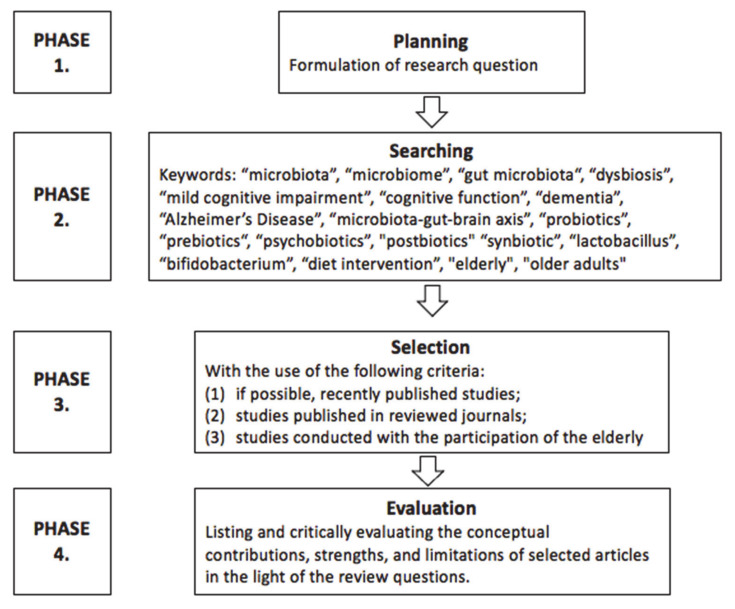
Diagram of review methodology.

**Figure 2 nutrients-13-02514-f002:**
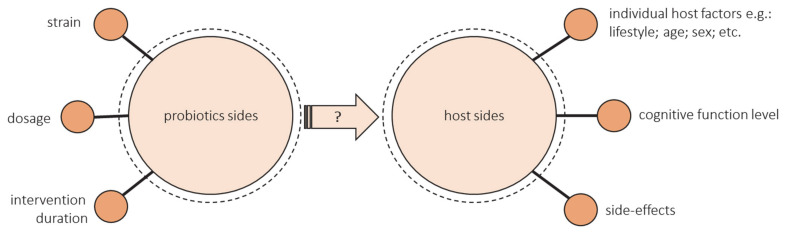
Unknowns that require clarification and parameters that need to be controlled from the probiotic and host sides in planning of intervention trials.

**Figure 3 nutrients-13-02514-f003:**
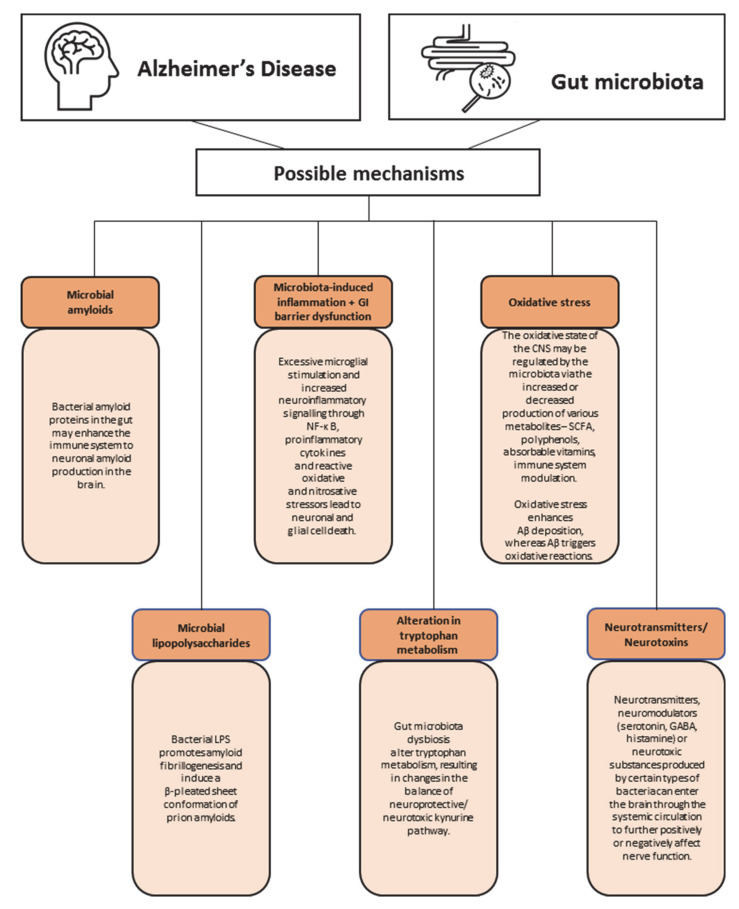
Possible mechanisms behind gut microbiota and Alzheimer’s Disease (AD). Abbreviations: CNS, central nervous system; GABA, γ-aminobutyric acid; SCFA, short-chain fatty acids.

**Table 1 nutrients-13-02514-t001:** Glossary of terms.

Terms	Definitions
“*Dysbiosis*”	is defined as a loss of beneficial microbes, expansion of pathobionts or potentially harmful microorganisms, and a loss of microbial diversity [20].
“*Microbiota*”	is defined as the microbial taxon (bacteria, archaea, or lower eukaryotes) associated with humans health and disease [7].
“*Microbiome*”	is defined as the catalog of microorganisms, their genomes, and the surrounding environmental condition [7], but we are lacking a clear commonly agreed definition [21].
“*Postbiotics*”	are defined as “preparation of inanimate microorganisms and/or their components that confers a health benefit on the host” [22] they are also known as “non-viable probiotics”, “ghost probiotics”, “paraprobiotics”, and “inactivated probiotics”, they have also ability to deliver health benefits if consumed with prebiotics or foods throughout their shelf life [23].
“*Prebiotics”*	defined as “a selectively fermented ingredient that allows specific changes, both in the composition and/or activity in the gastrointestinal microflora that confers benefits upon host wellbeing and health” [24].
“*Probiotics*”	are defined as “live microorganisms which when administered in adequate amounts confer a health benefit on the host” [25].
“*Psychobiotics*”	are defined as live bacteria which, when ingested in adequate amounts, produces a mental health benefits [26].
“*Synbiotic*”	contains combined both a probiotics and prebiotics that work synergistically and have beneficially affect the host, and this effect is higher than that of the probiotic alone [27].

**Table 2 nutrients-13-02514-t002:** Changes in gut microbiota composition in various forms of cognitive impairment.

Phyla	Class	Order	Family	Genus	Species	Cognition			References
						MCI	D	AD	
Firmicutes	*Clostridia*	*Clostridiales*						↓	[52]
			*Clostridiaceae*			↓		↓	[52]
				*Clostridium*				↓	[49]
		*Eubacteriales*	*Ruminococcaceae*					↓ ↓	[52] [49]
				*Ruminococcus*				↓	[52]
			*Lachnospiraceae*	- unclassified	*Lachnospiraceae bacterium* NK4A136	↓	↓ ↓	↓	[52] [51]
				*Blautia*		↓		↓ ↑	[52] [49]
			*Eubacteriaceae*	*Eubacterium*	*E. rectale*		↓ ↓	↑ ↑	[51] [50]
	*Erysipelotrichia*	*Erysipelotrichales*	*Turicibacteraceae*	*Turicibacter*				↓	[49]
	*Negativicutes*	*Veillonellales*	*Veillonellaceae*			↑			[52]
				*Dialister*				↓	[49]
		*Selenomonadales*	*Acidaminococcaceae*	*Phascolarctobacterium*				↑	[49]
	*Bacilli*	*Lactibacillales*	*Lactobacillaceae*	*Lactobacillus*			↑		[48]
		*Staphylococcales*	*Gemellaceae*	*Gemella*				↑	[49]
Bacteroidetes	*Bacteroidia*	*Bacteroidales*	*Bacteroidaceae*			↑ ↑			[52] [49]
				*Bacteroides*		↑ ↑	↓	↑ ↑ ↑	[47] [48] [49] [52]
					*B. fragilis*			↓	[50]
			*Rikenellaceae*	*Alistipes*				↑	[49]
Actinobacteria	*Actinomycetia*	*Bifidobacteriales*	*Bifidobacteriaceae*	*Bifidobacterium*			↑	↓	[47] [49]
	*Coriobacteriia*	*Eggerthellales*	*Eggerthellaceae*	*Adlercreutzia*				↓	[49]
Proteobacteria	*Gammaproteobacteria*	*Enterobacteriales*	*Enterobacteriaceae*					↑	[52]
				*Escherichia/Shigella*				↑	[50]
	*Deltaproteobacteria*	*Desulfovibrionales*	*Desulfovibrionaceae*	*Bilophila*				↑	[49]

Abbreviations: AD, Alzheimer’s Disease; D, dementia; MCI, Mild Cognitive Impairment.

**Table 3 nutrients-13-02514-t003:** Methodological issues of studies about microbiome composition in cognition impairment—case controls.

Country [References]	N (% Women) Cases	N (% Women) Controls	Age Cases	Age Controls	Cognitive Function Assessment/Diagnosis	Cognitive Impairment	Microbiome Assessment
Japan [49]	94 (85)	34 (49)	77, 74–82	76, 68–80	MMSE/CDR	Dementia	Fecal samples, bacterial 16 s rRNA Sequencing
Japan [50]	61 (54)	21 (48)	77, 73–81	69, 61–76	MMSE/CDR	MCI	Fecal samples, bacterial 16 s rRNA Sequencing
USA [51]	25 (72)	25 (68)	69 ± 7	71 ± 7	CDR	AD	Fecal samples, bacterial 16 s rRNA Sequencing
Austria [53]	23 (65)	18 (61)	88	75	MMSE	Dementia	Fecal samples, bacterial 16 s rRNA Sequencing
China [54]	32 (56) aMCI 33 (42) AD	32 (50)	70 ± 11 aMCI 75 ± 11 AD	77 ± 9	MMSE, CDR, MoCA	aMCI AD	Fecal samples, bacterial 16 s rRNA Sequencing
Italy [52]	40 (50) Amy+ 33 (52) Amy-	10 (60)	71 ± 7 Amy+ 70 ± 7 Amy-	68 ± 8	MMSE	AD (Amy+ Amy-)	Fecal samples, qPCR Assay, StepOnePlus instrument

Abbreviations: AD, Alzheimer’s Disease; aMCI, Amnestic MCI; Amy+, brain amyloidosis present; Amy-, brain amyloidosis absence; CDR, Clinical Dementia Rating; MCI, Mild Cognitive Impairment; MMSE, Mini-Mental State Examination; MoCA, Montreal Cognitive Assessment.

**Table 4 nutrients-13-02514-t004:** Summary of the evidence (from meta-analysis and systematic reviews) deriving from human studies aimed at evaluating the impact of nutrients on cognitive function or gut microbiota.

Nutrients	Type of Study/Impact on Cognitive Function	Type of Study/Impact on Gut Microbiota
proteins	**Observational studies**/no statistically significant differences in protein intake levels in AD patients and healthy controls [73]	*No meta-analysis and systematic reviews were identified.*
fats	**Prospective studies**/significantly higher risk of AD and dementia development in patients with higher SFA, while the results regarding unsaturated fatty acids intake were not statistically significant [74,75] RCT/no protective effects of omega-3 supplements in the prevention of cognitive decline, which is contradictory with previous findings [76]	**RCT, observational studies (cross-sectional)**/higher intake of fat and SFA was associated with poorer gut microbiota diversity, diets high in MUFA were associated with decreased total bacterial number, while whereas high PUFA intake had no effect on gut microbiota diversity and richness [77]
fiber	*No meta-analysis and systematic reviews were identified.*	**RCT**/dietary fiber resulted in higher abundance of *Bifidobacterium and Lactobacillus* spp. but does not affect α-diversity [78]
polyphenols	**RCT**/no definitive confirmation of the preventive effect of polyphenols on cognitive disorders [79] nor of neuroprotective effect of polyphenols supplementation in aging adults [80].	*No meta-analysis and systematic reviews were identified..*
vitamin B	**RCT**/no effect of B vitamins on cognitive function in older adults with or without cognitive impairment[81]	*No meta-analysis and systematic reviews were identified.*
vitamin D	**Observational (cross-sectional and longitudinal cohort) studies**/low vitamin D status is related to poorer cognition [82] and increased risk of cognitive impairment [83]. **Interventional studies with control group**/no significant effect of vitamin D supplementation on cognition [82]	**RCT, interventional and observational studies**/vitamin D influences the composition of the gastrointestinal microbiome (varied and limited studies) [84]
antioxidant vitamin	**Case–control studies**/significantly lower plasma levels of α-carotene, β-carotene, lycopene, lutein, vitamin A, C, and E in AD patients [85] **RCT**/no good evidence that supplementation can preserve cognitive function, prevent dementia [86]	*No meta-analysis and systematic reviews were identified.*

Abbreviations: AD, Alzheimer’s Disease; MUFA, monounsaturated fatty acids; PUFA, polyunsaturated fatty acids; RCT, randomized controlled trials; SFA, saturated fatty acids.

**Table 5 nutrients-13-02514-t005:** Methodological issues of randomized controlled trials (RCT) studies about probiotic, prebiotic, and synbiotic intervention on cognition function.

Intervention	Cognitive Impairment	Control Group		Study Group		Cognitive Function Assessment/Diagnosis	Microbiome Assessment	Other Assessed Parameters	Country [References]
		N	Age	N	Age				
PROBIOTICS									
*L. casei* Shirota 65 mL milk drink 6.5 × 10^9^ CFU 3 wk.	healthy	66	61.8	66	61.8	Memory (Wechsler Memory Scale) Retrieval from long-term memory Verbal fluency	no	eating-associated behaviour NART	UK [91]
*L. helveticus*-fermented milk drink 190 g/day 8 wk.	healthy	29	57.8 ± 5.9	31	58.5 ± 6.5	RBANS	no	POMS	Japan [92]
*L. helveticus*-fermented milk 500 mg, 1000 mg, 200 mg 12 wk.	healthy	10	64.5 ±4.8	10 7 9	64.5 ± 2.2 64.4 ± 4.5 66.6 ± 5.0	neuropsychological test battery (DST, VLT, SRT)	no	cognitive fatigue PSS GDS-SF BDNF WBV	Korea [93]
*B. bifidum*, *B. longum* 1 × 10^9^ CFU 12 wk.	healthy	26	72.0	27	71.1	CERAD-K	yes	BDNF	Korea [94]
*B. longum, B. infantis*, *B. breve*, *B. breve* 1.25 × 10^10^ CFU 12 wk.	healthy	18	70.9 ± 3.2	20	69.9 ± 3.0	MoCA-J Flanker task	no	PHQ-9, GAD-7, MNA, energy intake, BMI, height, weight, blood pressure, bowel movement characteristics	Japan [95]
*L. plantarum* ≥ 1.25 × 10^10^ CFU/day fermented soybean 2 capsules, once a day 12 wk.	MCI	50	69.2 ± 7.0	50	68.0 ± 5.1	CNT VLT ACPT DST	yes	BDNF, height, weight, blood pressure and pulse rate, complete blood cell count and blood parameters	South Korea [98]
*B. breve* > 1.0 × 10^10^ CFU/day 2 capsules 12 wk.	MCI	60	61.6 ± 6.4	61	61.5 ± 6.8	MMSE RBANS	no	hs-CRP haematological and biological blood parameters	Japan [99]
*L. fermentum*, *L. plantarum*, *L. acidophilus*, *B. lactis*, *B. longum*, *B. bifidum* 3 × 10^9^ CFU/g of each 2 capsules, once a day 12 wk.	AD	23	80.6 ± 1.8	25	79.7 ± 1.7	TYM	no	TAC, GSH, MDA, Il-6, Il-10, TNF-a, NO, 8-OHdG, weight, BMI	Iran [100]
*L. acidophilus*, *L. casei*, *L. fermentum B. bifidum*, 2 × 10^9^ CFU/g of each 200 mL probiotic milk/day 12 wk.	AD	30	82 ± 1.7	30	77.7 ± 2.6	MMSE	no	TAC, GSH, hs-CRP, MDA, NO, HOMA-B, HOMA-IR, QUICK, FPG, TG, TC, LDL, HDL, VLDL, TC/HDL, insulin, weight, BMI dietary intakes (3-day food records)	Iran [101]
*L. acidophilus*, *B. bifidum*, *B. longum* 2 × 10^9^ CFU/g each for probiotic capsule + selenium 200 ng/day 12 wk.	AD	52	78.8 ± 10.2	27	76.2 ± 8.1	MMSE	no	TAC, GSH, hsCRP, insulin, HOMA-IR, QUICKI, TG, NO, FPG, MDA, TC, LDL, VLDL, HDL, TC/HDL, Gene expression related to inflammation, insulin and lipid metabolism BMI, weight, dietary intakes (3-day food records)	Iran [102]
PREBIOTICS									
*Darmocare Pre^®^* (inulin + fructooligosaccharides) 13 wk.	non-demented	22	73.4 ± 1.8	28	74.2 ± 1.6	MMSE	no	Frailty Functional Impairment Sleep quality Blood analysis and haemogram	Spain [103]
SYNBIOTIC									
*L. paracasei*, *L. ramnosus*, *L. acidophilus*, *B. lactis* (10^8^–10^9^ CFU of each) + fructooligosaccharide twice a day 24 wk.	apparently healthy	24	77.0 ± 1.3	25	67.9 ± 4.5	MMSE	no	GDS-15, % of body fat, IL-6, TNF-α, IL-10, DAO, IFABP, LPS	Brazil [104]

Abbreviations: 8-OHdG, serum high sensitivity 8-hydroxy-2′-deoxyguanosine; ACPT, auditory continuous performance test; AD, Alzheimer’s Disease; BDNF, Brain-derived neurotrophic factor; CERAD-K, Consortium to Establish a Registry for Alzheimer’s Disease, The Korean version; CFU, colony-forming units; DAO, serum diamine-oxidase; DST, digit span test; DW2009, Lactobacillus plantarum C29-fermented soybean; FPG, fasting plasma glucose; GAD-7, Generalized Anxiety Disorder-7; GDS-SF, Geriatric Depression Scale, short form; GSH, total glutathione; HOMA-B, homeostatic model assessment for B-cell function; HOMA-IR, homeostasis model of assessment-insulin resistance; hs-CRP, high-sensitivity C-reactive protein; IFABP, intestinal fatty acid binding protein; IL-10, interleukin 10; LPS, lipopolysaccharide; MCI, Mild Cognitive Impairment; MDA, malondialdehyde; MMSE, Mini-Mental State Examination; MNA, Mini Nutritional Assessment; MoCA-J, Japanese version of the Montreal Cognitive Assessment instrument; NART, National Adult Reading Test; NO, nitric oxide; PHQ-9, Patient Health Questionnaire-9; POMS, The Short Form Of The Profile Of Mood States; PSS, perceived stress scale; QUICKI, quantitative insulin sensitivity check index; RBANS, Repeatable Battery for the Assessment of Neuropsychological Status; SRT, story recall test; TAC, total antioxidant capacity; TC, total cholesterol; TG, triglycerides; TNF-α, tumor necrosis factor α; VLT, verbal learning test; WBV, frictional force of blood flow in vessels.

## Data Availability

Not applicable.

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
