# Peer review of "Gut Microbiota, Probiotic Interventions, and Cognitive Function in the Elderly: A Review of Current Knowledge"

_nutrients, 2021, doi:10.3390/nu13082514_

Round 1

Reviewer 1 Report

This paper is a clear, complete and detailed literature review.
Here are some brief comments:

Line 92: It would have been interesting to use the word "postbiotics" as a research as well

Table 1: "Paraprobiotics" Postbiotic substances can also be produced as a result of taking prebiotics. Please clarify the concept of postbiotics.

Line 167- 169: This step is very interesting, you can provide more details?

Line 582: Could you further clarify the antioxidant effect of probiotics in the pathology of AD also in relation to the differences with the symbitoic molecules?

Reviewer 2 Report

In the manuscript ID- nutrients-1304003 titled “Gut microbiota, probiotic interventions, and cognitive function in the elderly: a review of current knowledge” by Agata BiaÅ‚ecka-DÄ™bek and colleagues. They have reported that the gut microbiota of cognitively healthy and impaired elderly people may differ in the diversity and abundance of individual taxes, but specific taxes cannot be identified. However, some tendencies to changing the Firmicutes/Bacteroidetes ratio can be identified. Currently, clinical trials involving probiotics, prebiotics, and synbiotics supplementation have shown that there are premises for the claim that these factors can improve cognitive functions, however, there is no single intervention beneficial to the elderly population. More reliable evidence from large-scale, long-period RCT is needed. I have few concerns regarding the present manuscript.

-The topic that the authors have investigated is interesting and novel in the field. In the introduction, they need new information for the definitions of microbiota and the relationship between aging and cognitive functions.

-The topic about research search needs dates, limitations, and numbers about the papers found in every step.

-Scopus and Cochrane are missing as databases in the present study

-What is the idea with the topic about health microbiota, in my opinion, the information from here maybe could be the new introduction.

-The next sections 4, and 5 are the best for me

-The tables are difficult to follow, maybe the authors need to summarise the information here, especially in table 5.

-Why the font letter is gray in some cases, maybe is for my computer only, please check

-Some species' names are incorrect, Bifidobacteria is not italic, maybe if the authors use Bifidobacterium yes.

-Please summarizes table 4 with more detailed information

-The authors have the idea to draw/made any figure for the present review

-In general, the manuscript is interesting and well-designed, which needs a revision and a final connecting part, maybe add more information in the conclusions with bullet points about the further directions in the field.

Round 2

Reviewer 2 Report

Thank you to the authors for considering my previous comments, I appreciated that. Concerning the present manuscript, in my opinion, the authors have an improved manuscript and a better view of their results, thanks for taking into account all my previous concerns.